# Core Bacterial Taxa Determine Formation of Forage Yield in Fertilized Soil

**DOI:** 10.3390/microorganisms12081679

**Published:** 2024-08-15

**Authors:** Xiangtao Wang, Ningning Zhao, Wencheng Li, Xin Pu, Peng Xu, Puchang Wang

**Affiliations:** 1School of Life Sciences, Guizhou Normal University, Guiyang 550025, China; wangxt@xza.edu.cn; 2State Key Laboratory of Herbage Improvement and Grassland Agro-Ecosystems, Center for Grassland Microbiome, College of Pastoral, Agriculture Science and Technology, Lanzhou University, Lanzhou 730000, China; 3Qiangtang Alpine Grassland Ecosystem Research Station, Tibet Agricultural and Animal Husbandry University, Nyingchi 860000, China; 4School of Ecology and Enviroment, Tibet University, Lhasa 850000, China; 5College of Animal Science, Tibet Agricultural and Animal Husbandry University, Nyingchi 860000, China; 6State Key Laboratory of Microbial Technology, Shandong University, Qingdao 266237, China

**Keywords:** organic manure, chemical N fertilization, soil core bacterial taxa, soil nutrient cycling, forage yield

## Abstract

Understanding the roles of core bacterial taxa in forage production is crucial for developing sustainable fertilization practices that enhance the soil bacteria and forage yield. This study aims to investigate the impact of different fertilization regimes on soil bacterial community structure and function, with a particular focus on the role of core bacterial taxa in contributing to soil nutrient content and enhancing forage yield. Field experiments and high-throughput sequencing techniques were used to analyze the soil bacterial community structure and function under various fertilization regimes, including six treatments, control with no amendment (CK), double the standard rate of organic manure (T01), the standard rate of organic manure with nitrogen input equal to T04 (T02), half the standard rate of inorganic fertilizer plus half the standard rate of organic manure (T03), the standard rate of inorganic fertilizer reflecting local practice (T04), and double the standard rate of inorganic fertilizer (T05). The results demonstrated that organic manure treatments, particularly T01, significantly increased the forage yield and the diversity of core bacterial taxa. Core taxa from the Actinomycetota, Alphaproteobacteria, and Gammaproteobacteria classes were crucial in enhancing the soil nutrient content, directly correlating with forage yield. Fertilization significantly influenced functions relating to carbon and nitrogen cycling, with core taxa playing central roles. The diversity of core microbiota and soil nutrient levels were key determinants of forage yield variations across treatments. These findings underscore the critical role of core bacterial taxa in agroecosystem productivity and advocate for their consideration in fertilization strategies to optimize forage yield, supporting the shift towards sustainable agricultural practices.

## 1. Introduction

Soil bacteria are essential for nutrient cycling, decomposition, and primary production [1,2]; however, the ecological roles of specific bacterial taxa, particularly core taxa, are often overlooked in favor of a broader focus on overall bacterial community functions [3]. Core bacterial taxa play crucial roles in various soil environments by decomposing organic matter and releasing essential nutrients such as nitrogen, phosphorus, and potassium [4,5]. For example, nitrogen-fixing bacteria convert atmospheric nitrogen into ammonia, making it available for plant uptake, while phosphate-solubilizing bacteria release phosphorus from insoluble compounds [3,6]. These core bacteria also produce growth-promoting substances like auxins, gibberellins, and enzymes that enhance plant development by aiding root growth and nutrient absorption [3,7,8]. Their abundance and activity can be sensitive to changes in soil conditions and agricultural practices, such as fertilizer application, which can significantly impact their ability to contribute to nutrient cycling and plant health [9]. The research shows that beneficial core bacteria in the rhizosphere are linked to improved soil health indicators, including increased soil organic carbon and total nitrogen levels, thereby enhancing nutrient mineralization and forage yield [10,11]. Despite these important functions, the specific roles of core and noncore bacterial taxa are often underexplored, with research frequently concentrating on broader community-level functions. Understanding these specific contributions is crucial for developing sustainable agricultural practices, as managing core bacterial taxa through targeted interventions can optimize their beneficial functions, promoting soil vitality, plant health, and sustainable forage yield.

Modern agriculture relies heavily on inorganic nitrogen (N) fertilizers, such as ammonium nitrate and urea, to boost forage yield by providing readily available nitrogen to plants [12,13,14]. However, the excessive use of these fertilizers negatively impacts soil bacteria, reducing bacterial diversity and overall soil health [15,16]. Inorganic nitrogen fertilizers disrupt the balance of soil microbial communities, leading to a decline in beneficial bacteria crucial for nutrient cycling and plant growth [17]. Additionally, the overuse of these fertilizers can result in soil salinization and reduced soil fertility due to salt accumulation [18,19]. Nitrate leaching into groundwater causes water pollution and poses health risks to humans and animals [20]. The evidence indicates that organic fertilization improves soil properties, increases beneficial bacterial species, and promotes bacterial functional diversity, ultimately helping to maintain crop productivity [21,22]. Regulating soil bacteria through optimized fertilization practices presents a sustainable pathway for forage production [23]. By promoting the growth of beneficial soil bacteria, optimized fertilization sustainably enhances plant growth and forage yield.

*Elymus nutans* Griseb., a perennial forage grass vital to Tibet’s agriculture and animal husbandry, thrives under the Tibetan Plateau’s extreme climate, offering indispensable sustenance for livestock [24,25]. However, the research on how fertilization affects its associated core bacterial taxa is scarce. A deeper understanding of these effects could enhance forage production and soil health in the region. This grass species, crucial for nutrition during the growing season, also combats soil erosion with its robust root system. While the overall soil bacterial community composition and its links to soil attributes and plant growth have been studied [25,26], the specific roles of core and noncore bacterial taxa remain underexplored. These taxa’s ecological functions are key to sustainable forage practices that support soil vitality and plant development. The research void centers on these taxa’s contributions to forage yield and soil wellness, a knowledge gap critical for advancing forage production sustainability.

Hence, we conducted a field fertilization experiment to investigate the role of core bacterial taxa in determining the forage yield in fertilized soil. We hypothesized that core bacterial taxa are more closely associated with forage yield than noncore bacteria and that core bacteria affect forage yield by mediating soil organic and inorganic nitrogen. Specifically, we aimed to achieve the following: (1) to explore the effect of fertilization on core and noncore bacterial taxa and their functions; (2) to quantify the contributions of these taxa to functional characteristics, soil nutrients, and forage yield; and (3) to establish the interactions among fertilization, core and noncore bacterial taxa, functional characteristics, soil nutrients, and forage yield. We employed advanced amplicon sequencing techniques to analyze soil bacterial communities and identify core and noncore taxa. By examining the effects of different fertilization regimes on these bacterial communities, we elucidated the potential mechanisms through which soil bacteria influence forage yield. Our findings have significant implications for sustainable forage production, providing a theoretical foundation for optimizing fertilization practices to enhance soil health and plant growth.

## 2. Method and Materials

### 2.1. Site Description

The experiment was conducted at the Grassland Science Experiment Base of the Tibet Agriculture and Animal Husbandry College (94°20′39″ E, 29°40′32″ N). The site has a temperate monsoon climate, an altitude of 2983 m, an annual average temperature of 8.6 °C, and annual average precipitation of 650 mm, mostly occurring from June to September. The mean temperature of the warmest and coldest months was 15.6 °C and 0.2 °C, respectively. The extreme maximum and minimum temperatures were 30.20 °C and −15.3 °C, respectively. The annual average effective accumulated temperature (≥10 °C) was above 2000 °C, and the frost-free period was longer than 180 days. The soil at the site was classified as loam according to the FAO system, with an average depth of approximately 60 cm. The soil physicochemical properties before experiment in the experimental region were as follows (means ± standard deviations, *n* = 9): organic carbon (OC), 5.46 (g/kg); total nitrogen (TN), 1.13 ± 0.32 (g/kg); total phosphorus (TP), 0.78 (g/kg); and available phosphorus (AP), 21.81 (mg/kg). Seeds of *Elymus nutans* used for this study were collected from the Grassland Science Experimental Base of the Tibet Agriculture and Animal Husbandry College in September 2020. These seeds are a domesticated variant of the wild *Elymus nutans* originally procured in the Nagqu region.

### 2.2. Experimental Design

The field experiment, which was performed in September 2021, used a completely randomized design with six treatments and six replicates per treatment. The treatments were as follows: (1) CK, no fertilization; (2) T01, double the standard rate of organic manure; (3) T02, standard rate of organic manure with N input equal to N100 (i.e., 5848 kg of manure ha^−1^y^−1^); (4) T03, half the standard rate of inorganic fertilizer plus half the standard rate of organic manure; (5) T04, standard rate of inorganic fertilizer reflecting local practice (i.e., 47% urea, 100 kg N ha^−2^ y^−1^); and (6) T05, double the standard rate of inorganic fertilizer. The experiment consisted of 36 plots, each measuring 20 m^−2^ (4 m × 5 m). To hydrologically isolate each plot, trenches 1 m deep were dug around each plot and lined with plastic film and PVC sheets. All treatments received P and K as 75 kg ha^−2^ y^−1^ phosphorus pentoxide (46% P_2_O_5_) and 45 kg ha^−2^ y^−1^ potassium chloride (60% KCl), applied as a base fertilizer at sowing. Organic manure and total mineral fertilizer P and K were applied once at the time of sowing, while total mineral fertilizer N was applied twice per year: half was applied at the time of sowing and the other half was applied at the jointing stage. The organic manure was sheet manure from the dairy industry nearby and it was composted by regular turning (3–4 times) over a 4-month period before application. Sheep manure contained 293.23 g/kg of total carbon, 17.10 g/kg of total nitrogen, 3.79 g/kg of total phosphorus, 5.28 g/kg of total potassium (measured on a dry matter basis), and 60% of water content.

### 2.3. Sampling and Analysis

Soil samples for bacterial community analysis were collected from each plot post-harvest on June 20 and August 20, 2022. We gathered samples from a 0–20 cm depth using five cores (5 cm diameter) and an “S” sampling method, then quartered the soil to obtain about 1 kg samples. These were immediately sealed in sterile bags and transported on ice to the lab within 24 h. In total, 72 soil samples (6 treatments × 6 replicates × 2 time points) were sieved (2 mm) to remove debris and divided into two portions. One was stored at 4 °C for physicochemical analysis, and the other was frozen at −80 °C for DNA extraction.

We determined soil physicochemical properties using standard methods [27]. Briefly, we measured soil organic carbon (OC), available phosphorus (AP), pH, an ammonium (NH_4_^+^-N), and nitrate N (NO_3_^−^-N) in air-dried soil. This was followed by H_2_SO_4_–K_2_Cr_2_O_7_ oxidation, the Kjeldahl and Olsen methods, colorimetrically by the molybdate-ascorbic acid method (UV-1800; Shimadzu, Kyoto, Japan) and a flow injection autoanalyzer (AutAnalyel; Bran + Luebbe GmbH, Norderstedt, Germany). The dissolved organic carbon (DOC) and dissolved organic nitrogen (DON) were stored at 4 °C and measured using a TOC analyzer (Liqui TOC II; Elementar, Germany).

### 2.4. Microbial DNA Extraction, Sequencing, Bioinformatics Analysis

DNA was extracted from 0.25 g of homogenized soil samples utilizing the FastDNA^®^ SPIN Kit according to the manufacturer’s instructions. The DNA concentrations were quantified with a UV-Vis spectrophotometer. For amplification of the 16S rRNA v3-v4 region, we employed the 338F/806R primer set [27]. Subsequently, the PCR products underwent purification using the Qiagen Gel Extraction Kit. The final step involved paired-end sequencing on the Illumina MiSeq 300 platform. The soil bacterial data set is deposited in the Genome Sequence Archive at the BIG Data Center, Beijing Institute of Genomics [28,29], Chinese Academy of Sciences, under the accession number PRJCA002453 and is publicly accessible at http://bigd.big.ac.cn/gsa (accessed on 31 December 2023).

The analysis of sequenced samples was conducted using QIIME2 [30] and USEARCH v11.0 [31]. Initially, we removed primer and low-quality sequences (Q < 30), then merged the paired-end sequences using the -fastq_mergepairs command. Following this, we applied -fastq_filter for quality control and -fastx_uniques for sequence de-duplication. We identified exact sequence variants (ESVs) via the Unoise3 algorithm [32] and assigned taxa using the Silva v138 database within QIIME2. Ultimately, we obtained 21,974 ESVs in the bacterial domain from 72 samples, normalized to 28,313 sequences per sample for further analysis.

### 2.5. Statistical Analysis

We identified core bacterial taxa based on the following two criteria: (1) taxa with a relative abundance in the top 10% across all samples, and (2) taxa present in over 90% of soil samples [3,5,33]. Taxa not meeting these criteria were classified as noncore. To assess the ecological roles of both core and noncore taxa under various fertilization treatments, we first calculated all pairwise sparse correlations for compositional (SparCC) correlations between bacterial nodes using the FastSpar algorithm, employing 100 bootstraps and 100 permutations to control the false discovery rate [34,35]. Prior to constructing the co-occurrence networks for bacterial communities, we removed ASVs with a relative abundance of less than 0.001% to minimize the impact of the rare ASVs [36]. We considered correlation coefficients with an absolute R value above 0.80 and *p*-values below 0.01 as significant [35]. The resulting networks were visualized using Gephi (v 10.1; https://gephi.org/, accessed on 20 July 2024).

We assessed the impact of fertilization regimes on soil properties, microbial diversity, and community composition using one-way ANOVA and Tukey’s test, with *p*-values under 0.05 indicating significance. Principal coordinate analysis (PCoA) based on Bray–Curtis dissimilarity measured the differences in microbial communities across fertilization treatments, complemented by PERMANOVA for significance testing. Multiple regression analysis determined the correlations between the microbial taxa and soil nutrients, highlighting the key taxa in nutrient cycling. Their relative importance was gauged through hierarchical partitioning. Finally, to understand the influences of microbial taxa and soil nutrients on forage yield, we employed partial least squares path modeling (PLS-PM). This statistical method is particularly useful for demonstrating cause-and-effect relationships among observed and latent variables [37,38]. We utilized the R package “plspm” (v. 3.3.3) to estimate the path coefficients and coefficients of determination (R^2^) in our path model, applying 1000 bootstraps for validation. The models’ overall predictive power was evaluated using the goodness of fit (GOF) statistic, with a GOF value greater than 0.7 considered acceptable [39]. Models were constructed using the function “inner plot” in the “plspm” package [37]. Path coefficients illustrate the strength and direction of linear relationships between variables. This methodology enabled us to thoroughly examine the relationships between microbial communities, soil nutrients, and forage yield.

All analyses were conducted in the R statistical environment (v 4.3.0; https://www.r-project.org/, accessed on 20 July 2024) using “vegan” [40], “ggplot2” [41], “picante” [42], and “rdacca.hp” [43] packages.

## 3. Results

### 3.1. Forage Yield and Soil Properties under Different Fertilization Regimes

Significant differences in forage yield were observed across all samples among the various fertilization treatments (Figure 1). Both T01 and T02 treatments, which received organic manure, showed significantly higher forage yields compared to the mineral and control treatments. Forage yield increased with higher manure input rates, peaking in the T01 treatment. There was no significant difference between the T04 and T05 treatments. Similarly, the analysis of soil chemical properties indicated that, except for pH, there were no significant variations among the different fertilization treatments (Table 1). Fertilization significantly increased the OC, NH_4_^+^-N, NO_3_^−^-N, DON, DOC, and AP contents compared to the control treatment, with the highest values observed in the T01 treatment.

### 3.2. Identifications and Diversity Pattern of Core and Rare Bacterial Taxa

To identify the core microbiota, we selected the most abundant and ubiquitous ASVs across all soil samples. Overall, the abundant taxa accounted for a significantly lower proportion of ASVs (mean 16.38%) but a larger proportion of average relative abundance (mean 73.35%) in each sample compared to the noncore taxa (means 83.62% and 26.65%, respectively). The PCoA revealed that the structure of the core bacterial sub-community was more like the overall community than the noncore sub-community (Appendix A). This was confirmed by Mantel tests (core vs. whole: r = 0.9986, *p* < 0.01; noncore vs. whole: r = 0.5537, *p* < 0.01). These core taxa were mainly classified into Actinomycetota, Alphaproteobacteria, Gammaproteobacteria, Bacteroidota, and Thermoleophilia (Figure 2). The PCoA results showed that the β-diversity of both core and noncore bacteria under different treatments formed distinct clusters in the ordination space, with significant differences among treatments (Figure 3a–f, *p* < 0.01). The T01 treatments had the highest core bacterial α-diversity (Shannon diversity and species richness) among the treatments, while the lowest was observed in CK (Figure 4a,b). Similarly, organic manure treatments significantly increased the noncore bacterial α-diversity, although there was no significant difference between T01 and T02 (Appendix A). However, the variance in Shannon diversity induced by fertilization regimes was significantly higher in the core compared to the noncore (Appendix A).

### 3.3. Ecological Role of Core Bacterial Taxa

We further investigated the ecological roles of core microbiota in maintaining bacterial taxa connections. Metacommunity co-occurrence networks, based on correlations, comprised 2243 nodes (ASVs) and 3426 edges (Figure 5a). Notably, 7.13% of the nodes were core taxa, capturing 98.35% of the edges (including core–core and core–noncore links). The average degree and betweenness of different subcommunities were significantly higher (*p* < 0.01; Wilcoxon rank sum tests) for core taxa compared to noncore taxa (Figure 5b and Appendix A). A random forest model revealed that core bacterial genera significantly influence soil nutrient properties, explaining 80.81%, 84.95%, 83.34%, 50.85%, 78.70%, and 83.31% of the variations in OC, NH_4_^+^-N, NO_3_^−^-N, AP, DON, and DOC, respectively, with most genera showing positive correlations. Additionally, we evaluated the relationships between specific core taxa (at the genus level) and soil nutrient levels across all fertilization treatments (Figure 6). Changes in OC, NH_4_^+^-N, NO_3_^−^-N, AP, DOC, and DON levels were related to the abundances of Vicinamibacteraceae, Pseudonocardia, and Cellulomonas, while variations in NO_3_^−^-N and DON content were more closely correlated with Rhizobium abundance. Furthermore, OC content was associated with Rhizobium abundance.

Fertilization regimes significantly altered the relative abundance of chitinolysis and aerobic ammonia oxidation in bacteria (Figure 7 and Appendix A, ANOVA, *p* < 0.01). Compared to inorganic N treatments, organic manure treatment increased the abundance of chitinolysis, aerobic ammonia oxidation, and nitrification by 65.33%, 25.12%, and 32.55%, respectively. The T01 treatment had more functions related to C and N cycling than inorganic nitrogen treatments. Additionally, there were significant correlations between core bacterial taxa and C and N cycling functions, while noncore bacterial taxa were only correlated with N cycling functions (Appendix A).

### 3.4. Possible Drivers of Soil Nutrients Cycling and Forage Yield

Multiple regression analysis explored the correlation between bacterial taxa diversity and forage yield, while hierarchical partitioning quantified the relative abundance of these indices on forage yield variation across fertilization treatments. The Shannon diversity, species richness, and β-diversity of the core microbiota explained 25.97% of the variations in forage yield (Table 2), whereas the α- and β-diversity of the noncore microbiota explained only 8.77%. We further constructed a PLS-PM to assess the relationships among fertilization regimes, core bacterial taxa, functional characteristics related to C and N cycling, available nitrogen, OC, DOC, and wheat yield (Figure 8). The factors in the PLS-PM explained 79% of the variation in forage yield. Core bacterial taxa shaped by inorganic and organic fertilization indirectly affected forage yield by influencing functional characteristics (*p* < 0.01). The functional characteristics positively impacted soil nutrient status (*p* < 0.001). These results were also supported by linear regression analysis, indicating that the core taxa diversity was significantly positively correlated with forage yield (Appendix A).

## 4. Discussion

### 4.1. Fertilization Treatments Significantly Changed the Forage Yield and Bacterial Communities

The results of our study indicate that organic manure applications (T01 and T02 treatments) significantly enhance the forage yield compared to the mineral and control treatments, with the highest yields observed in the T01 treatment. This outcome aligns with previous findings that organic amendments improve soil fertility and plant growth by enhancing nutrient availability and soil structure [44,45]. The increase in soil organic carbon, ammonium nitrogen, nitrate nitrogen, dissolved organic nitrogen, dissolved organic carbon, and available phosphorus observed in our study further supports the positive impact of organic fertilizers on soil nutrient dynamics (Table 1). These enhanced soil chemical properties likely contribute to the improved forage yield, as higher nutrient availability can directly support plant growth and productivity.

Our findings on the bacterial community dynamics reveal distinct responses to different fertilization treatments. The core microbiota, primarily composed of Actinomycetota, Alphaproteobacteria, Gammaproteobacteria, Bacteroidota, and Thermoleophilia, showed a high degree of resilience to external disturbances such as fertilization, as evidenced by the significant similarity between the core bacterial sub-community and the overall community. This resilience suggests that core microbiota play a crucial role in maintaining soil ecosystem stability and functionality, consistent with previous studies indicating that core taxa are often resource acquisition strategists with broad ecological niches [5,46]. In contrast, noncore microbiota exhibited greater variability in response to fertilization, with significant differences in α-diversity and community structure across treatments. This variability can be attributed to the noncore taxa’s lower abundance and higher sensitivity to environmental changes [10,11], aligning with the concept that noncore taxa, often stress-tolerant strategists, rapidly respond to changes in resource availability and other external factors [47]. The enhanced α-diversity of both core and noncore bacterial communities in the T01 treatment highlights the beneficial effects of organic manure on microbial diversity. This increase in diversity is crucial for soil health, as diverse microbial communities are associated with enhanced ecosystem multifunctionality, including nutrient cycling and organic matter decomposition [48,49]. The significant increase in noncore bacterial diversity under organic manure treatments, without a corresponding increase in core diversity, suggests that organic inputs particularly benefit less dominant, yet functionally important, microbial taxa. These findings support the idea that organic amendments not only improve soil fertility but also promote a more diverse and functionally robust microbial community [21,44,50]. The principal coordinate analysis (PCoA) and Mantel tests further illustrate the distinct clustering of bacterial communities under different treatments, with the T01 treatment promoting the most distinct and diverse community structures. This pattern underscores the profound impact of organic amendments on shaping soil microbial communities, potentially leading to improved soil health and plant productivity. The correlation between high microbial diversity and improved soil chemical properties, such as increased OC, NH_4_^+^-N, NO_3_^−^-N, DON, DOC, and AP contents, reinforces the integral role of diverse microbial communities in soil nutrient cycling and overall ecosystem functionality.

### 4.2. Organic Manure Addition Enhances Forage Yield by Promoting Core Bacterial Taxa and Influencing Soil C and N Cycling

The burgeoning interest in soil microbiomes as crucial contributors to agricultural sustainability is warranted due to their significant impact on ecological equilibrium and productivity [51,52]. This research elucidates the essential functions of core microbiota in shaping bacterial communities, influencing soil nutrient dynamics, and boosting forage yield under different fertilization practices, particularly with organic fertilizers. Recognized for their ability to modify soil microbiomes, organic fertilizers specifically enhance core microbial taxa that are vital for nutrient cycling. Our results are consistent with Chen et al. [53], who observed that organic amendments significantly modified the composition and functionality of soil microbial communities, thereby increasing their biodiversity and metabolic proficiency. In this study, core taxa such as Vicinamibacteraceae, Pseudonocardia, and Cellulomonas not only demonstrated greater abundance but also displayed enhanced connectivity within the microbial networks compared to noncore taxa. These taxa form integral nodes in dense co-occurrence networks, regulating community structure and resilience. The targeted modification of core groups, particularly those involved in nitrogen and carbon cycles, reveals the mechanisms through which organic fertilizers affect soil health. Enhancements in functions like chitinolysis, aerobic ammonia oxidation, and nitrification following organic manure treatments suggest a strengthened biochemical framework for nitrogen and carbon cycling. This supports and expands upon the work of Schmidt et al. [54], who noted increased enzymatic activities linked to nitrogen cycling in organically managed soils.

Furthermore, the stability and functionality of microbial communities are intricately linked. Our research indicates that core microbiota, through their extensive connections within the microbial network, sustain a stable community structure essential for ongoing soil fertility [3,47,55]. The significant correlations between the diversity of core taxa and various soil nutrient properties (e.g., organic carbon, ammonium–nitrogen, nitrate–nitrogen) underline the pivotal role these groups play in maintaining ecological balance and enhancing nutrient availability. These insights align with Jiang et al.’s study [56], who explored how microbial community stability could facilitate functional resilience and ecosystem services in agricultural soils. The notable betweenness centrality and degree of core taxa in our findings underscore their role as keystone species within the network, fostering interactions and cohesions essential for functional efficacy [55,57]. Employing advanced statistical models, our study decoded the intricate relationships between microbial community structure and function, and forage yield. The partial least squares path modeling (PLS-PM) approach revealed that the influence of core taxa on yield was mediated by their effects on soil nutrient dynamics and microbial community functions. This indirect influence is significant, suggesting that enhancing microbial community structure—especially the diversity of core taxa—may improve forage yields [10,58]. This hypothesis is supported by Ye et al. [46], who identified a similar correlation between microbial community composition and crop yield in cereal systems. The substantial variation in forage yield explained by core microbiota diversity (25.97%) in our study underscores the critical role these organisms play in agricultural systems. Hence, fertilization serves as a biological catalyst that activates the core bacterial taxa to augment productivity in agricultural ecosystems. Future research should delve deeper into fertilization strategies to fully leverage the ecological functions of core bacterial taxa. Consequently, core bacterial taxa, often underestimated, have been shown to enhance forage yield by boosting soil nutrient supply and modifying functions related to carbon and nitrogen cycling.

## 5. Conclusions

In conclusion, our study demonstrates that different fertilization regimes significantly affect forage yield, soil nutrients, and core bacterial taxa, with organic manure treatments showing particularly beneficial effects. These treatments enhance forage yield and improve soil nutrient properties by promoting key bacterial groups, such as Actinomycetota and Pseudomonadota, which are essential for nutrient cycling and forage production. Based on these findings, we recommend that both farmers and policymakers prioritize the use of organic fertilizers wherever possible. This approach not only increases agricultural productivity by enhancing forage yield but also fosters beneficial microbial communities that support sustainable nutrient cycling. By incorporating organic manure into soil management practices, we can achieve improved soil fertility and sustainability, providing a holistic solution that benefits both agricultural output and environmental health. Future research should focus on elucidating the specific roles of core microbiota in nutrient cycling and forage production. Such understanding will refine fertilization practices and support the development of sustainable agricultural strategies, ensuring long-term soil health and productivity.

## Figures and Tables

**Figure 1 microorganisms-12-01679-f001:**
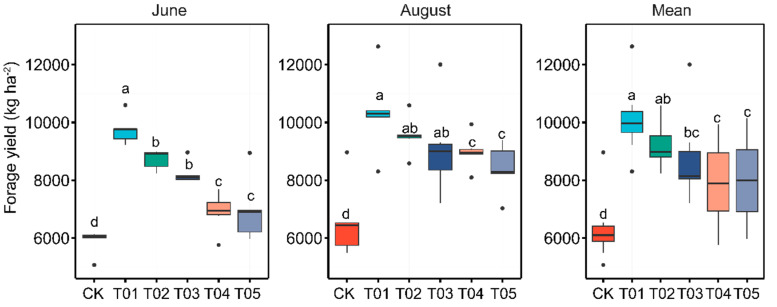
Variations in forage yield were analyzed across different fertilization regimes and sampling times. Significant differences among fertilization treatments at each sampling time were identified using Duncan’s multiple range test (*p* < 0.05), as denoted by lowercase letters in the results. CK, control with no amendment addition; T01, double the standard rate of organic manure; T02, standard rate of organic manure with N input equal to T04; T03, half the standard rate of inorganic fertilizer plus half the standard rate of organic manure; T04, standard rate of inorganic fertilizer reflecting local practice; and T05, double the standard rate of inorganic fertilizer.

**Figure 2 microorganisms-12-01679-f002:**
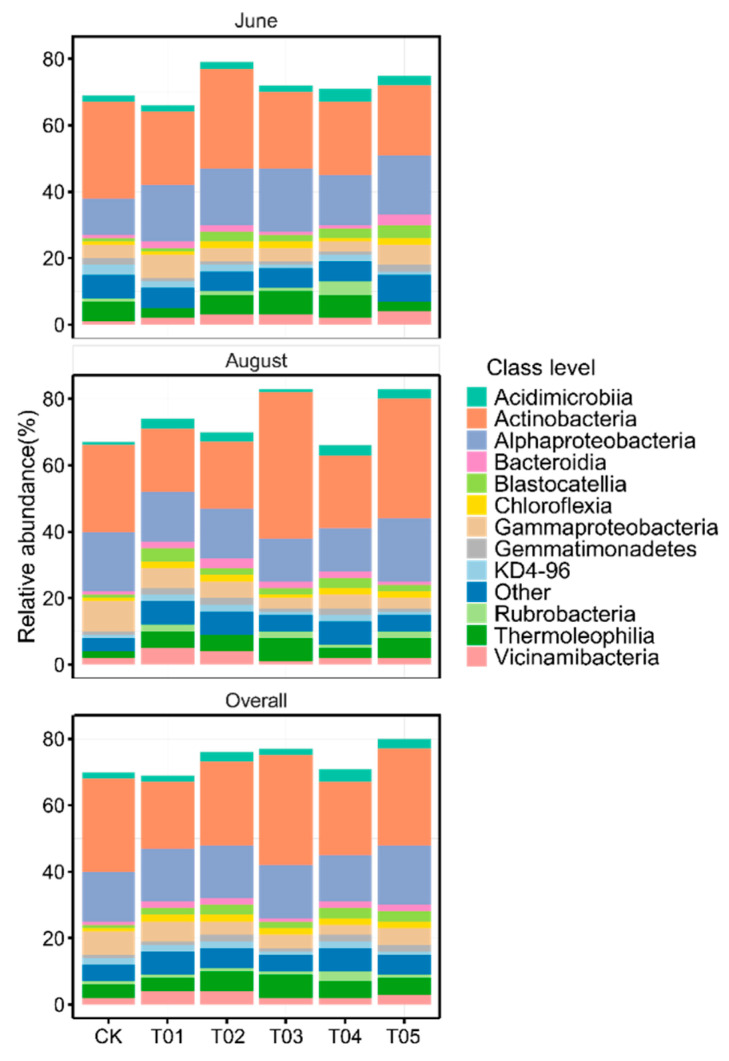
The relative abundance of core bacterial taxa at class level across varied fertilization regimes and sampling times. CK, control with no amendment addition; T01, double the standard rate of organic manure; T02, standard rate of organic manure with N input equal to T04; T03, half the standard rate of inorganic fertilizer plus half the standard rate of organic manure; T04, standard rate of inorganic fertilizer reflecting local practice; and T05, double the standard rate of inorganic fertilizer.

**Figure 3 microorganisms-12-01679-f003:**
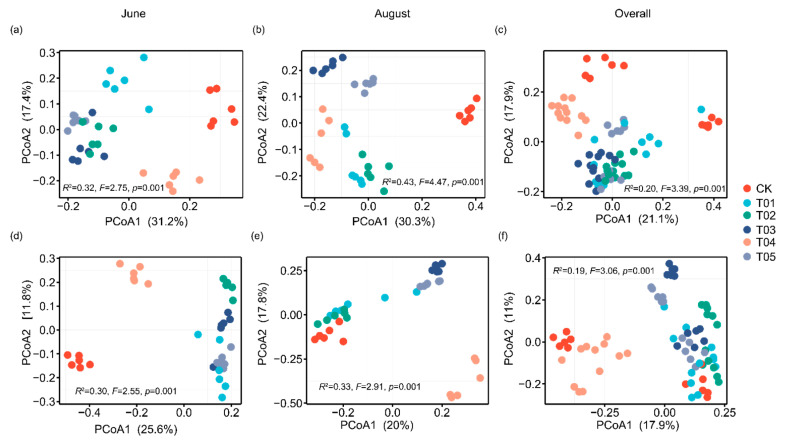
Principal coordinate analysis delineated the clustering of core (**a**–**c**) and noncore (**d**–**f**) bacterial communities under different fertilization treatments and sampling times. Bacterial community variation was assessed using PERMANOVA (999 permutations) based on Bray–Curtis dissimilarity. CK, control with no amendment addition; T01, double the standard rate of organic manure; T02, standard rate of organic manure with N input equal to T04; T03, half the standard rate of inorganic fertilizer plus half the standard rate of organic manure; T04, standard rate of inorganic fertilizer reflecting local practice; and T05, double the standard rate of inorganic fertilizer.

**Figure 4 microorganisms-12-01679-f004:**
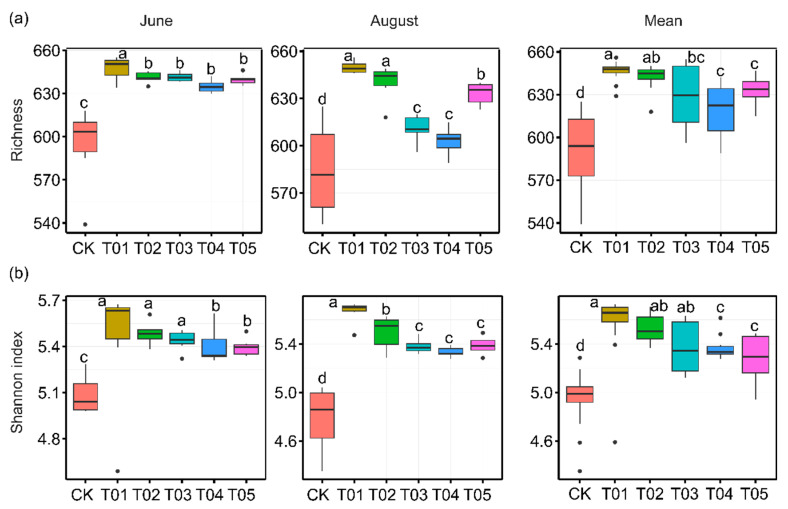
The alpha diversity of core bacterial taxa was assessed across various fertilization regimes and sampling times. Significant differences in alpha diversity among fertilization treatments were determined by Duncan’s multiple range test (*p* < 0.05). CK, control with no amendment addition; T01, double the standard rate of organic manure; T02, standard rate of organic manure with N input equal to T04; T03, half the standard rate of inorganic fertilizer plus half the standard rate of organic manure; T04, standard rate of inorganic fertilizer reflecting local practice; and T05, double the standard rate of inorganic fertilizer.

**Figure 5 microorganisms-12-01679-f005:**
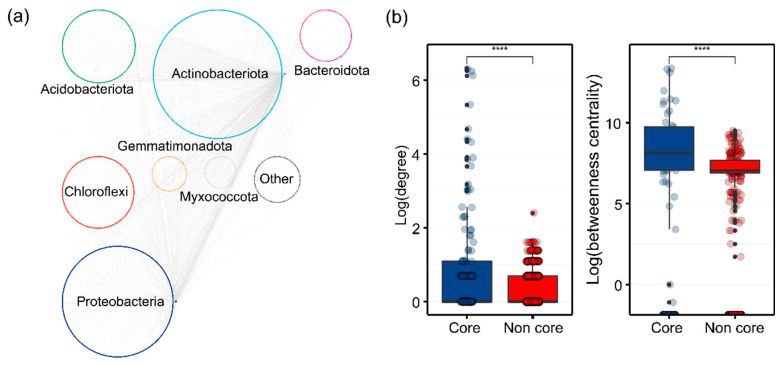
Metacommunity co-occurrence networks of bacterial taxa in fertilized soils were analyzed using SparCC. The analysis revealed distinct node-level topological features between core and noncore bacterial taxa, with statistical significance determined by Wilcoxon rank sum tests (****, *p* < 0.001). Networks were color-coded by dominant taxa at the phylum level, with connections representing strong, significant correlations (correlation coefficient > 0.9, *p* < 0.01).

**Figure 6 microorganisms-12-01679-f006:**
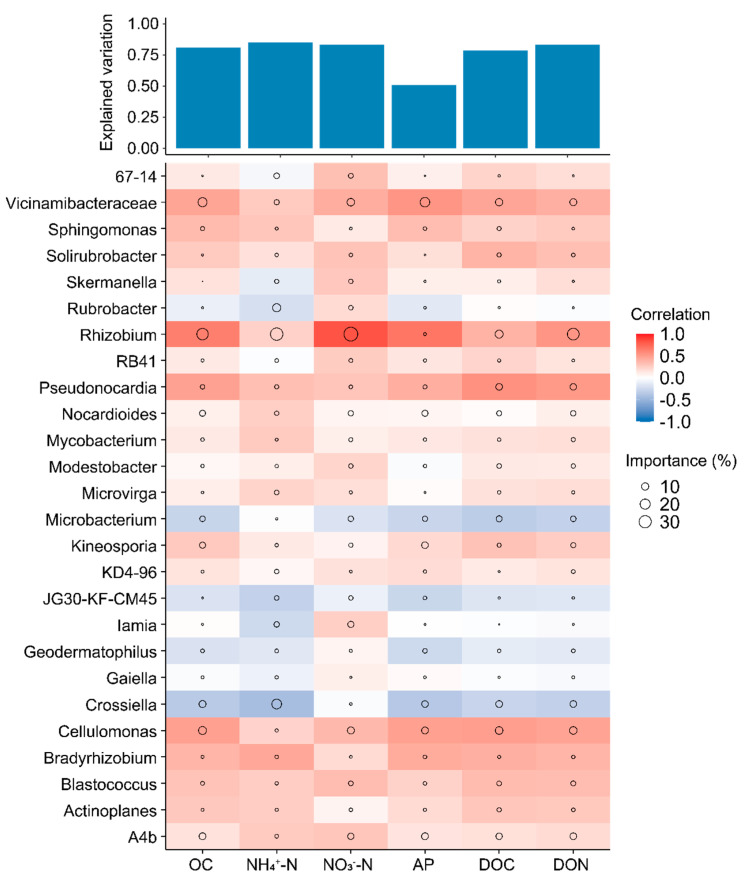
The potential biological contributions of core microbiota at the genus level to soil nutrient properties were evaluated using multiple regression analysis and hierarchical partitioning. Circle sizes in the visual representations correlate with the importance of variables, while colors indicate Spearman correlations between microbial genera and soil nutrients including organic carbon (OC), ammonium-nitrogen (NH_4_^+^-N), nitrate-nitrogen (NO_3_^−^-N), Olsen-phosphorus (AP), dissolved organic carbon (DOC), and dissolved organic nitrogen (DON).

**Figure 7 microorganisms-12-01679-f007:**
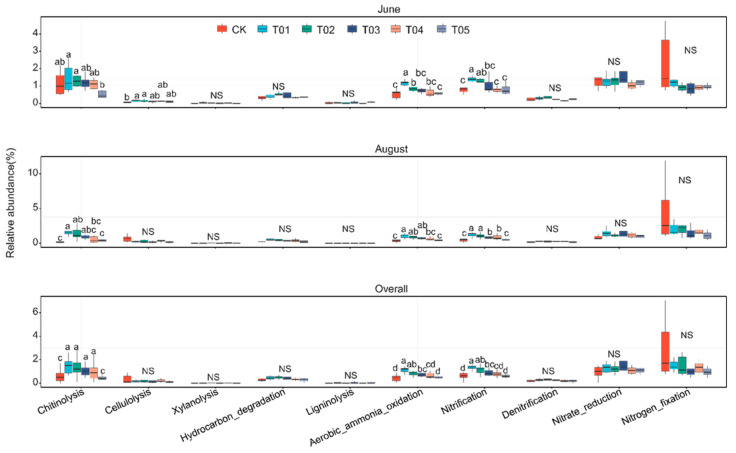
Functional characteristics related to carbon and nitrogen cycling, as predicted by FAPROTAX, showed significant differences across fertilization treatments at the same sampling times (Duncan’s multiple range test, *p* < 0.05). Non-significance among treatments is noted as NS. CK, control with no amendment addition; T01, double the standard rate of organic manure; T02, standard rate of organic manure with N input equal to T04; T03, half the standard rate of inorganic fertilizer plus half the standard rate of organic manure; T04, standard rate of inorganic fertilizer reflecting local practice; and T05, double the standard rate of inorganic fertilizer.

**Figure 8 microorganisms-12-01679-f008:**
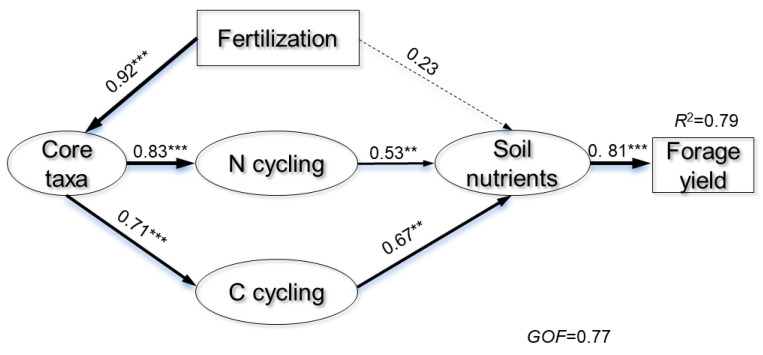
The partial least squares path modeling (PLS-PM) was utilized to evaluate the direct and indirect impacts of core bacterial taxa, functional characteristics related to carbon and nitrogen cycling, and soil nutrients on forage yield. Path widths correspond to the magnitude of path coefficients, with solid arrows indicating significant relationships and dashed arrows indicating non-significant relationships. The R2 value indicates the proportion of variance explained, with significance levels noted as ** *p* < 0.01, and *** *p* < 0.001, respectively.

**Table 1 microorganisms-12-01679-t001:** Alterations in soil physicochemical properties under various fertilizer regimes and at different sampling times are presented as means with standard errors (SEs). Distinct lowercase letters indicate significant differences among fertilizer regimes within a single sampling period at *p* < 0.05. CK, control with no amendment addition; T01, double the standard rate of organic manure; T02, standard rate of organic manure with N input equal to T04; T03, half the standard rate of inorganic fertilizer plus half the standard rate of organic manure; T04, standard rate of inorganic fertilizer reflecting local practice; and T05, double the standard rate of inorganic fertilizer.

Sampling Times	Fertilization	OC (g/kg)	NH_4_^+^-N (mg/kg)	NO_3_^−^-N (mg/kg)	DOC (mg/kg)	DON (mg/kg)	pH	AP (mg/kg)
June	CK	5.75 ± 0.09 e	11.51 ± 0.54 c	1.65 ± 0.08 e	34.47 ± 3.06 d	27.04 ± 1.76 e	8.43 ± 0.11 a	22.10 ± 1.16 d
	T01	20.06 ± 0.53 a	16.32 ± 0.17 a	7.27 ± 0.37 a	180.75 ± 4.85 a	114.80 ± 3.62 a	8.41 ± 0.11 a	58.08 ± 1.64 a
	T02	14.27 ± 0.31 b	13.75 ± 0.19 b	5.79 ± 0.18 b	152.38 ± 19.04 a	92.41 ± 4.43 b	8.23 ± 0.08 a	45.50 ± 1.76 b
	T03	12.37 ± 0.35 c	13.17 ± 0.26 b	4.90 ± 0.26 c	106.54 ± 12.49 b	81.58 ± 3.42 c	8.39 ± 0.06 a	33.52 ± 1.06 c
	T04	9.74 ± 0.38 d	3.21 ± 0.08 d	6.7 ± 0.30 a	75.38 ± 6.78 c	47.48 ± 0.60 d	8.22 ± 0.09 a	30.57 ± 2.90 c
	T05	10.47 ± 0.49 d	11.55 ± 0.54 c	3.87 ± 0.02 d	91.39 ± 2.30 bc	49.48 ± 3.24 d	8.33 ± 0.08 a	31.29 ± 1.40 c
August	CK	5.93 ± 0.31e	11.23 ± 0.70 c	1.65 ± 0.06 e	36.28 ± 1.87 d	27.72 ± 2.33 d	8.15 ± 0.08 a	26.52 ± 1.32 e
	T01	20.47 ± 0.19 a	16.10 ± 0.11 a	6.22 ± 0.35 a	183.55 ± 9.52 a	115.63 ± 4.87 a	8.38 ± 0.06 a	60.76 ± 0.79 a
	T02	14.82 ± 0.28 b	13.27 ± 0.40 b	5.68 ± 0.30 ab	154.80 ± 18.72 b	96.46 ± 3.41 b	8.26 ± 0.12 a	46.15 ± 0.92 b
	T03	12.48 ± 0.29 c	13.41 ± 0.21 b	5.34 ± 0.35 bc	146.19 ± 3.99 b	88.07 ± 3.57 b	8.38 ± 0.09 a	34.53 ± 0.77 c
	T04	10.14 ± 0.22 d	11.13 ± 0.38 c	3.86 ± 0.04 d	75.21 ± 4.02 c	49.89 ± 1.68 c	8.23 ± 0.11 a	30.74 ± 1.30 cd
	T05	10.55 ± 0.24 d	12.58 ± 0.09 b	4.76 ± 0.13 c	93.76 ± 7.21 c	56.15 ± 2.37 c	8.32 ± 0.09 a	28.21 ± 2.38 de
Mean	CK	5.84 ± 0.22 e	11.37 ± 0.6 c	1.65 ± 0.07 d	35.38 ± 2.45 e	27.38 ± 1.98 e	8.29 ± 0.11 a	24.31 ± 1.51 e
	T01	20.26 ± 0.39 a	16.21 ± 0.14 a	6.75 ± 0.41 a	182.15 ± 7.23 a	115.22 ± 4.1 a	8.4 ± 0.08 a	59.42 ± 1.36 a
	T02	14.55 ± 0.31 b	13.51 ± 0.32 b	5.74 ± 0.24 b	153.59 ± 18.01 b	94.44 ± 3.87 b	8.25 ± 0.10 a	45.82 ± 1.35 b
	T03	12.42 ± 0.31 c	13.29 ± 0.23 b	5.12 ± 0.31 b	126.36 ± 12.23 c	84.83 ± 3.61 c	8.39 ± 0.08 a	34.02 ± 0.91 c
	T04	9.94 ± 0.31 d	7.17 ± 1.71 d	5.28 ± 0.64 b	75.30 ± 5.32 d	48.69 ± 1.31 d	8.23 ± 0.10 a	30.65 ± 2.14 d
	T05	10.51 ± 0.37 d	12.06 ± 0.43 bc	4.32 ± 0.21 c	92.58 ± 5.13 d	52.82 ± 3.06 d	8.33 ± 0.08 a	29.75 ± 1.97 d

**Table 2 microorganisms-12-01679-t002:** Variation in forage yield explained by bacterial diversity indices of core and noncore bacterial subcommunities in regression models.

		Forage Yield (%)
Core		
	Alpha-Richness	8.63
	Alpha-Shannon	10.55
	Beta-PCoA1	3.77
	Beta-PCoA2	3.02
	Total	25.97
Noncore		
	Alpha-Richness	2.33
	Alpha-Shannon	1.59
	Beta-PCoA1	4.22
	Beta-PCoA2	0.63
	Total	8.77

## Data Availability

The original contributions presented in the study are included in the article/Appendix A, further inquiries can be directed to the corresponding author.

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
