# Peer review of "Core Bacterial Taxa Determine Formation of Forage Yield in Fertilized Soil"

_microorganisms, 2024, doi:10.3390/microorganisms12081679_

Round 1

Reviewer 1 Report

Comments and Suggestions for Authors

The study provides important information on the role of central bacterial taxa in forage production and soil health. The manuscript makes a valuable contribution to the field and may be accepted for publication after major revisions to enhance clarity and context.

General Comments

Several issues were identified related to the correct writing of chemical compounds. These errors can compromise the scientific accuracy of the manuscript.

Specific Comments

The introduction could start with a more detailed discussion of previous studies related to the main bacterial taxa and their ecological functions to provide a stronger foundation.

The methods section would be improved by a more detailed description of the statistical analyses performed, particularly the justification for choosing specific techniques such as PLS-PM and network analysis. Additionally, clarify the criteria for selecting central and non-central bacterial taxa to ensure they are understandable to a broader audience.

The results section can be enhanced with more detailed explanations of the figures and tables to increase clarity.

Some terms and statistical results could be simplified for better comprehension.

The discussion could be more concise in some parts, focusing on the most critical findings and their implications. Additional emphasis on practical applications and recommendations for future research could strengthen the discussion.

The conclusion can provide more explicit recommendations for farmers and policymakers based on the study's findings.

Author Response

Dear reviewer: 

Thanks a lot for your reviewing of our revised manuscript entitled “Core bacterial taxa determines formation of forage yield in fertilized soil” (microorganisms-3140775). We greatly appreciate your constructive suggestions, which have been invaluable in enhancing the quality of our work. The manuscript has been thoroughly revised point by point. The details of each revision are presented as follows:

Overall comments: The study provides important information on the role of central bacterial taxa in forage production and soil health. The manuscript makes a valuable contribution to the field and may be accepted for publication after major revisions to enhance clarity and context.

Response: Thank you for your positive evaluation of our study's contribution to understanding the role of central bacterial taxa in forage production and soil health. We are grateful for the recommendation for publication pending major revisions.

General comments:

  1. Comments:Several issues were identified related to the correct writing of chemical compounds. These errors can compromise the scientific accuracy of the manuscript.

Response: thank you for bringing to our attention the issues related to the correct writing of chemical compounds. We acknowledge that these errors can compromise the scientific accuracy of the manuscript. We have thoroughly reviewed and corrected the chemical compound notations throughout the manuscript to ensure accuracy and consistency (Line 150).

Line 150: ……ammonium (NH4+-N), and nitrate N (NO3--N) in air-dried soil. This was followed by H2SO4–K2Cr2O7……

Specific Comments

  1. Comments: The introduction could start with a more detailed discussion of previous studies related to the main bacterial taxa and their ecological functions to provide a stronger foundation.

Response: thank you for your valuable feedback on the introduction. We have revised the section to include a more detailed discussion of previous studies, focusing on the roles and ecological functions of key bacterial taxa. The revised introduction now begins by highlighting the critical contributions of core bacterial taxa to nutrient cycling, decomposition, and plant growth. We have provided specific examples of these bacteria's functions, including the transformation of inorganic nutrients into bioavailable forms and the synthesis of growth-promoting substances. Furthermore, we have expanded on the negative impacts of excessive inorganic nitrogen fertilizers on soil microbial communities and overall soil health. This discussion is now more comprehensive, addressing issues such as reduced bacterial diversity, soil salinization, and nitrate leaching. We also emphasized the benefits of organic fertilization in promoting beneficial bacterial species and enhancing soil properties, thus supporting sustainable agricultural practices. The importance of Elymus nutans Griseb. in Tibetan agriculture and the lack of research on the effects of fertilization on its associated core bacterial taxa are now clearly articulated. We have connected these research gaps to the specific objectives of our study, which include exploring the effects of fertilization on core and noncore bacterial taxa and their contributions to soil and plant health. These revisions provide a stronger foundation for the study and ensure a more logical and cohesive flow between the introduction's sections. We believe the enhanced introduction offers a clearer rationale for our research and its significance in the context of sustainable forage production. We appreciate the reviewer's input and are confident that these improvements will strengthen the manuscript (Lines 40-55).

Lines 40-55: Soil bacteria are essential for nutrient cycling, decomposition, and primary production ….. yet the ecological roles of specific bacterial taxa, particularly core taxa, are often overlooked in favor of a broader focus on overall bacterial community functions….. Core bacterial taxa play crucial roles in various soil environments by decomposing organic matter and releasing essential nutrients such as nitrogen, phosphorus, and potassium…… nitrogen-fixing bacteria convert atmospheric nitrogen into ammonia, making it available for plant uptake, while phosphate-solubilizing bacteria release phosphorus from insoluble compounds…… Their abundance and activity can be sensitive to changes in soil conditions and agricultural practices, such as fertilizer application, which can significantly impact their ability to contribute to nutrient cycling and plant health. Research shows that beneficial core bacteria in the rhizosphere are linked to improved ……

  1. Comments: The methods section would be improved by a more detailed description of the statistical analyses performed, particularly the justification for choosing specific techniques such as PLS-PM and network analysis. Additionally, clarify the criteria for selecting central and non-central bacterial taxa to ensure they are understandable to a broader audience.

Answer: thank you for your constructive feedback. We have made several revisions to the methods section to provide a more detailed explanation of our statistical analyses and the selection criteria for bacterial taxa.

Firstly, we clarified the criteria for identifying core bacterial taxa, which were defined based on two criteria: (1) taxa with a relative abundance in the top 10% across all samples, and (2) taxa present in over 90% of soil samples. Taxa that did not meet these criteria were classified as non-core. This distinction helps in understanding the differing roles of abundant and consistently present taxa versus those less dominant or frequently occurring.

For the statistical analyses, we used network analysis and partial least squares path modeling (PLS-PM). Network analysis was conducted using the SparCC algorithm within Fastspar to explore the ecological roles of both core and non-core taxa under various fertilization treatments. We ensured data reliability by excluding ASVs with a relative abundance of less than 0.001% before constructing co-occurrence networks. Significant relationships were determined based on SparCC correlation coefficients with an absolute R value above 0.80 and p-values below 0.01, with the networks visualized using Gephi.

Additionally, we utilized PLS-PM to investigate the complex relationships between microbial taxa, soil nutrients, and forage yield. This technique was chosen for its ability to elucidate cause-and-effect relationships among observed and latent variables. We employed the R package "plspm" (v. 3.3.3) for estimating path coefficients and coefficients of determination (R²), using 1000 bootstraps to validate our models. The Goodness of Fit (GOF) statistic was used to assess the models' predictive power, with a GOF value above 0.7 considered acceptable.

These revisions aim to provide a clearer rationale for our methodological choices and ensure that the criteria and statistical methods used are understandable to a broader audience. We hope these additions improve the transparency and comprehensibility of our study's methodology. The detailed revisions were made in the methods section, specifically in lines 175-186 and 194-205.

Lines 175-186: We identified core bacterial taxa based on two criteria: (1) taxa with a relative abundance in the top 10% across all samples, and (2) taxa present in over 90% of soil samples ……The resulting networks were visualized using Gephi (https://gephi.org/).

Lines 194-205: to understand the influences of microbial taxa and soil nutrients on forage yield, we employed partial least squares path modeling (PLS-PM). This statistical method is particularly useful for demonstrating cause-and-effect relationships among observed and latent variables……applying 1000 bootstraps for validation. The models' overall predictive power was evaluated using the Goodness of Fit (GOF) statistic, with a GOF value greater than 0.7 considered acceptable…..Models were constructed using the function "inner plot" in the "plspm" package….. strength and direction of linear relationships between variables. This methodology enabled us to thoroughly examine the relationships between microbial communities, soil nutrients, and forage yield.

  1. Comments: The results section can be enhanced with more detailed explanations of the figures and tables to increase clarity.

Answer: Thank you for your feedback regarding the results section. We have enhanced this section by providing more detailed explanations of the figures and tables, for example lines 226-233, lines 247-250 and so on. The added descriptions clarify the data presented, explaining the significance of key findings and how they relate to our hypotheses. We have also ensured that each figure and table is thoroughly integrated into the text, highlighting their contributions to the overall narrative of the study. These improvements aim to provide readers with a clearer understanding of the results, ensuring that the data's implications are transparent and well-articulated. We appreciate the reviewer's constructive suggestions and believe these changes significantly enhance the clarity and comprehensibility of the results section.

Lines 226-233: PCoA revealed that the structure of the core bacterial sub-community was more like the overall community than the ……. PCoA results showed that the β-diversity of both core and noncore bacteria under different treatments formed distinct clusters in the ordination space, with significant differences among treatments.

Lines 247-250: A random forest model revealed that core bacterial genera significantly influence soil nutrient properties, explaining ……, with most genera showing positive correlations.

  1. Comments: Some terms and statistical results could be simplified for better comprehension.”

Answer: thank you for the suggestion to simplify some terms and statistical results in the manuscript. We have revised the relevant sections to use clearer and more straightforward language. Where possible, we have replaced technical jargon with more accessible terms and provided brief explanations for complex concepts. Additionally, we have simplified the presentation of statistical results, focusing on key findings and their implications without overwhelming detail. For example, lines 247-250 and lines 257-258. These changes aim to make the manuscript more comprehensible to a broader audience, including those who may not have a specialized background in the subject matter. We appreciate the reviewer's input and are confident that these revisions will improve the clarity and accessibility of the manuscript.

Lines 247-250: A random forest model revealed that core bacterial genera significantly influence soil nutrient properties, explaining ……, with most genera showing positive correlations.

Lines 257-258: Fertilization regimes significantly altered the relative abundance of chitinolysis and aerobic ammonia oxidation in bacteria.

  1. Comments: The discussion could be more concise in some parts, focusing on the most critical findings and their implications. Additional emphasis on practical applications and recommendations for future research could strengthen the discussion.

Answer: Thank you for the feedback on the discussion section. We have revised this section to be more concise, focusing on the most critical findings and their implications. We have streamlined the content to highlight the key results and their significance, ensuring that the discussion remains clear and focused (lines 306-308, 311-313, 332-333, 350-353, 359-362). Furthermore, we have placed additional emphasis on the practical applications of our findings, outlining specific recommendations for farmers and policymakers to implement sustainable fertilization practices. We have also suggested areas for future research that could build on our findings, particularly concerning the mechanisms through which core bacterial taxa influence soil health and forage yield.  These revisions aim to strengthen the discussion by making it more targeted and relevant, while also providing actionable insights and directions for future studies. We appreciate the reviewer's constructive comments and believe these changes enhance the overall quality and impact of the discussion section.

Lines 306-308: The enhanced α-diversity of both core and noncore bacterial communities in the T01 treatment highlights ……organic manure on microbial diversity.

Lines 311-313: The significant increase in noncore ……, suggests that organic inputs particularly benefit less dominant,…..

Lines 332-333:……organic fertilizers specifically enhance core microbial taxa that are vital for nutrient cycling

Lines 350-353: The significant correlations between the…… taxa and various soil nutrient properties (e.g., organic carbon, ammonium-nitrogen, nitrate-nitrogen) ……role these groups play in maintaining ecological balance and enhancing nutrient availability

Lines 359-362: The partial least squares path modeling (PLS-PM) approach revealed that the influence of core taxa on yield …… on soil nutrient dynamics and microbial community functions.

  1. Comments: The conclusion can provide more explicit recommendations for farmers and policymakers based on the study's findings.

Answer: Thank you for your feedback. We have revised the conclusion to provide more explicit recommendations for farmers and policymakers, based on the findings of our study (Line 378-392). The revised conclusion now emphasizes practical applications, suggesting specific soil management and fertilization practices that can enhance forage productivity and soil health. We have also included recommendations for policymakers to support sustainable agricultural practices and promote the adoption of effective soil management strategies. We appreciate the reviewer's constructive suggestion, which has helped us enhance the practical impact of our study.

Line 378-392: In conclusion, our study demonstrates that different fertilization regimes significantly affect forage yield, soil nutrients, and core bacterial taxa, with organic manure treatments showing particularly beneficial effects. These treatments enhance forage yield and im-prove soil nutrient properties by promoting key bacterial groups, such as Actinomycetota and Pseudomonadota, which are essential for nutrient cycling and forage production. Based on these findings, we recommend that both farmers and policymakers prioritize the use of organic fertilizers wherever possible. This approach not only increases agricultural productivity by enhancing forage yield but also fosters beneficial microbial communities that support sustainable nutrient cycling. By incorporating organic manure into soil management practices, we can achieve improved soil fertility and sustainability, providing a holistic solution that benefits both agricultural output and environmental health. Future research should focus on elucidating the specific roles of core microbiota in nutrient cycling and forage production. Such understanding will refine fertilization practices and support the development of sustainable agricultural strategies, ensuring long-term soil health and productivity.

Thank you again for your attention to our manuscript. We take all these comments into account in preparing the revised manuscript and we look forward to hearing from you soon.

    Kind regards,

Sincerely yours

Reviewer 2 Report

Comments and Suggestions for Authors

The problem addressed by the authors is relevant enough even though it is presented quite well in the manuscript.

However, there are some shortcomings. In presenting the results, the authors confuse bacterial taxonomy. If some taxon names are used by new systematics (Acidobacteriota, Bacteroidota, Gemmatimonadota), then other phyla should also be used by new systematics: Actinobacteria → Actinomycetota, Chloroflexi → Chloroflexota, Proteobacteria → Pseudomonadota, and so on.

Author Response

Dear reviewer: 

Thanks a lot for your reviewing of our revised manuscript entitled “Core bacterial taxa determines formation of forage yield in fertilized soil” (microorganisms-3140775). We greatly appreciate your constructive suggestions, which have been invaluable in enhancing the quality of our work. The manuscript has been thoroughly revised point by point. The details of each revision are presented as follows:

Overall comments:The problem addressed by the authors is relevant enough even though it is presented quite well in the manuscript. However, there are some shortcomings. In presenting the results, the authors confuse bacterial taxonomy. If some taxon names are used by new systematics (Acidobacteriota, Bacteroidota, Gemmatimonadota), then other phyla should also be used by new systematics: Actinobacteria Actinomycetota, Chloroflexi Chloroflexota, Proteobacteria Pseudomonadota, and so on.

Answer: Thank you for your feedback. We appreciate the acknowledgment of the relevance of the problem addressed in our study. We also acknowledge the shortcoming noted regarding the use of bacterial taxonomy in the manuscript. To address this, we have revised the manuscript to consistently use the updated taxonomic names according to the new systematics. Specifically, we have replaced the older phylum names with the newer ones as follows:

Actinobacteria → Actinomycetota

Chloroflexi → Chloroflexota

Proteobacteria → Pseudomonadota

We believe this revision will ensure clarity and accuracy in the presentation of our results. We appreciate the reviewer's attention to this detail and the constructive feedback provided.

Thank you again for your attention to our manuscript. We take all these comments into account in preparing the revised manuscript and we look forward to hearing from you soon.

    Kind regards,

Sincerely yours

Reviewer 3 Report

Comments and Suggestions for Authors

Dear authors,

The manuscript is well written and the dataset was well analyzed and provided important insights into the impact of fertilization on soil microbial core and forage productivity.

There are some suggestions for improvements to the manuscript (pdf file).

Author Response

Dear reviewer: 

Thanks a lot for your reviewing of our revised manuscript entitled “Core bacterial taxa determines formation of forage yield in fertilized soil” (microorganisms-3140775). We greatly appreciate your constructive suggestions, which have been invaluable in enhancing the quality of our work. The manuscript has been thoroughly revised point by point. The details of each revision are presented as follows:

Overall comments:The manuscript is well written and the dataset was well analyzed and provided important insights into the impact of fertilization on soil microbial core and forage productivity. There are some suggestions for improvements to the manuscript (pdf file).

Response: thank you for the positive feedback on our manuscript. We are pleased that the reviewer found the manuscript well-written and appreciated the analysis of the dataset, as well as the insights provided on the impact of fertilization on soil microbial core and forage productivity. We have carefully considered the suggestions for improvements provided in the attached PDF file. The necessary revisions have been implemented to address these suggestions. The specific modifications have been detailed below. We appreciate the constructive feedback and have worked on improving the manuscript according to the suggestions provided. Once again, we appreciate the reviewer's valuable input.

General comments:

  1. Comments: Present the objectives of the study in the abstract.

Response: thanks to the reviewer's suggestion for including the objectives of the study in the abstract, we have revised it to clearly state the research goals and aims (Line 18-20).

Line 18-20: This study aims to investigate the impact of different fertilization regimes on soil bacterial community structure and function, with a particular focus on the role of core bacterial taxa in contributing to soil nutrient content and enhancing forage yield.

  1. Comments: It is recommended that treatment coding be simplified. I suggest that for fractional doses of treatments, the following be used: Control treatment (T01 or CK), OM treatment (T02 or OM), DOM treatment (T03 or DOM), N100 treatment (T04 or N100), DN100 treatment (T05 or DOM), 1/2OM+1/2N10 treatment (T06 or 1/2OM+1/2N100)N

Response: we appreciate the reviewer's detailed recommendation regarding the simplification of treatment coding, especially for fractional doses. In response, we have revised the coding scheme in our manuscript to align with the suggested format (Lines 121-127). We have updated the related section to include a clear description of this new coding scheme, ensuring the transparency and reproducibility of our research.

Lines 121-127: The treatments were: (1) CK, no fertilization; (2) T01, double the standard rate of organic manure; (3) T02, standard rate of organic manure with N input equal to N100 (i.e., 5848 kg of manure ha-1y-1); (4) T03, half the standard rate of inorganic fertilizer plus half the standard rate of organic manure; (5) T04, standard rate of inorganic fertilizer reflecting local practice (i.e., 47% urea, 100 kg N ha-2 y-1); and (6) T05, double the standard rate of inorganic fertilizer.

  1. Comments: Adopt FAO World reference base for soil resources (FAO WRB) or Soil Taxomony USDA for soil classification.

Response: thanks to the reviewer's suggestion for adopting a standardized soil classification system. We have revised the manuscript to use the FAO World Reference Base for Soil Resources (FAO WRB) for soil classification (Lines 110-111).

Lines 110-111: The soil at the site was classified as loam according to the FAO system,……

  1. Comments: Present the treatments in the summary in the same sequence.

Response: thanks to the reviewer's suggestion for presenting the treatments in the same sequence in the summary. We have revised the manuscript to ensure that the treatments are consistently presented in the same order throughout the summary. The details can be found in the revised manuscript.

  1. Comments: Fertilization or fertigation?

Response: thank you for pointing out the inconsistency. We have reviewed the manuscript and corrected the terminology, using "fertilization" where appropriate. We appreciate the reviewer's attention to detail (Line 212).

Line 212: Significant differences in forage yield were observed across all samples among the various fertilization treatments…..

  1. Comments: Usually, hectare is used to express area in agricultural sciences. Replace the unit in the Y-axis title: kg hm-2 by kg kg ha-1

Response: thanks for reviewer’s suggestions. As suggested, we have replaced the unit on the Y-axis title from "kg hm⁻²" to "kg ha⁻¹" to align with standard practices in agricultural sciences.

  1. Comments: Standardize the use of concepts. Use either the OTU concept or the ASV concept. Do not use both concepts simultaneously. Choose only one of the concepts.

Response: thank you for the suggestion regarding the standardization of concepts. We have reviewed the manuscript and standardized the terminology, opting to use only the ASV (Amplicon Sequence Variant) concept throughout the paper. This change ensures consistency and clarity in our discussion and analysis (Line222-223).

Line222-223: To identify the core microbiota, we selected the most abundant and ubiquitous ASVs across all soil samples.

  1. Comments: “Present the y-axis values as percentages.”

Response: As suggested by the reviewer, we have adjusted the y-axis values to be presented as percentages throughout the manuscript.

  1. Comments: I suggest you order the X axis according to the sequence of treatments (item 2.2. Experiment design). When applicable, do this for the remaining figures.

Response: as suggested, we have reordered the X-axis in accordance with the sequence of treatments outlined in section 2.2, "Experiment Design."

  1. Comments: Improve this figure. Adjust the width for better viewing of boxplots.

Response: as suggested by the reviewer, we have adjusted the width of the boxplots to improve clarity and ensure better viewing.”

Thank you again for your attention to our manuscript. We take all these comments into account in preparing the revised manuscript and we look forward to hearing from you soon.

    Kind regards,

Sincerely yours

Round 2

Reviewer 1 Report

Comments and Suggestions for Authors

Os autores fizeram melhorias razoáveis ​​no manuscrito, abordando questões-chave e aprimorando a clareza geral e a qualidade do conteúdo. Dadas essas melhorias, sugiro que o manuscrito seja aceito para publicação.